# SE(3)-Equivariant Relational Rearrangement with Neural Descriptor Fields

**Anthony Simeonov**[*,1,2], **Yilun Du**[*,1], **Lin Yen-Chen**[1]
**Alberto Rodriguez**[3], **Leslie Pack Kaelbling**[1], **Tomás Lozano-Pérez**[1], **Pulkit Agrawal**[1,2]
Massachusetts Institute of Technology
[1]CSAIL, [2] Improbable AI Lab, [3]Department of Mechanical Engineering, [*]Equal Contribution

**Abstract:** We present a framework for specifying tasks involving spatial relations between objects using only ∼5-10 demonstrations and then executing such tasks given point cloud observations of a novel pair of objects in arbitrary initial poses. Our approach structures these rearrangement tasks by assigning a consistent local coordinate frame to the task-relevant object parts, localizing the corresponding coordinate frame on unseen object instances, and executing an action that brings these frames into alignment. We propose an optimization method that uses multiple Neural Descriptor Fields (NDFs) and a *single* annotated 3D keypoint to assign a set of consistent coordinate frames to the task-relevant object parts. We also propose an energy-based learning scheme to model the joint configuration of the objects that satisfies a desired relational task. We validate our pipeline on three multi-object rearrangement tasks in simulation and on a real robot. Results show that our method can infer relative transformations that satisfy the desired relation between novel objects in unseen initial poses using just a few demonstrations.

**Keywords:** Object Relations, Rearrangement, Manipulation, Neural Fields

## 1 Introduction

Many tasks we want robots to perform – e.g., stacking bowls and plates to declutter a table, putting objects together to build an assembly, and hanging mugs on a rack with hooks – involve rearranging objects relative to one another. Such tasks can be described in terms of spatial relations between *part* features of a set of objects, where a *local* coordinate frame is attached to the task-relevant part of the object, and the relation is achieved by transforming the objects to bring these coordinate frames into a specified alignment. For example, *hanging* a mug on a rack is a relation between the mug's handle and the rack's hook, while *stacking* a bowl on a mug involves aligning the bottom of the bowl with the top of the mug (see Fig. 1).

Specifying and solving tasks in this way requires the ability to (i) assign a *consistent* local coordinate frame to the *task-relevant* object parts, and (ii) *detect* the corresponding coordinate frames on new object instances. Prior works have demonstrated these capabilities using techniques such as supervised keypoint detection [1, 2], but the use of large task-specific datasets labeled by humans limits easy deployment for a wide diversity of tasks. Neural Descriptor Fields (NDFs) [3] have also been used to perform these abilities, with the added benefit of requiring just a small set (∼ 5-10) of task demonstrations. They achieve this by combining task-agnostic self-supervised pretraining and a few labeled examples of objects with consistent coordinate frames attached to their task-relevant parts, effectively performing few-shot learning for task-relevant part localization on new instances.

Despite these benefits, it can be tedious to label the relevant part of each demonstration object with a consistent pose (e.g., label the "handle" of each mug with a consistent orientation). Prior work [3] reduced this burden by permitting the user to label a *single* coordinate frame near the task-relevant part of a *known* secondary object, and use the demonstrations to associate the frame with the task-relevant part of each *unknown* object (e.g., label a frame on the "hook" of a *known* rack *once*, and associate

---

Project website: https://anthonysimeonov.github.io/r-ndf/

6th Conference on Robot Learning (CoRL 2022), Auckland, New Zealand.

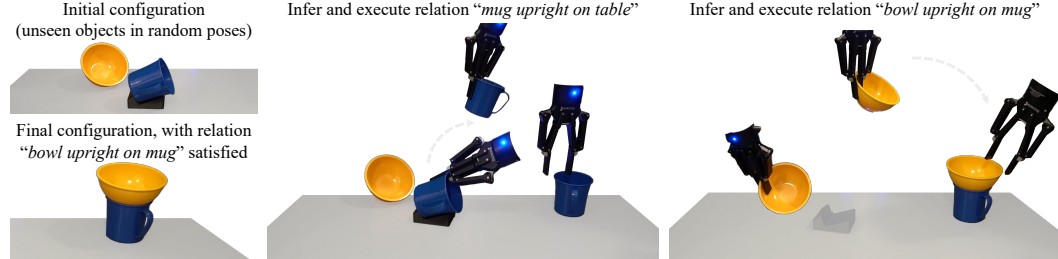

Initial configuration
(unseen objects in random poses)

Infer and execute relation "*mug upright on table*"

Infer and execute relation "*bowl upright on mug*"

Final configuration, with relation
"*bowl upright on mug*" satisfied

Figure 1: Given a point cloud of a pair of unseen objects in aritrary initial configurations (top left), Relational Neural Descriptor Fields (R-NDFs) obtain relative transformations that satisfy a relational task objective, such as "placing the mug upright on the table" (middle) and "stacking the bowl upright on top of the mug" (right). Our framework obtains these transformations by inferring the 6D pose of local coordinate frames at the task-relevant parts of the objects using just a small handful (∼5-10) of demonstrations of each relational task.

this frame with each mug's "handle" based on the final mug placements). This labeled frame also specifies the alignment target for test-time objects once their relevant parts are localized. While this enables generalization to unseen objects in diverse poses [3], assuming a known secondary object is limiting – for the *hanging* example, the system generalizes to scenarios with unseen mugs, but fails with both an unseen mug *and* a rack with a new shape. In this paper, we address this limitation. In particular, we present Relational Neural Descriptor Fields (R-NDFs), a framework, using ∼ 5-10 demonstrations, that takes as input 3D point clouds of a *pair* of unseen objects in arbitrary initial poses and outputs a relative transformation between them that satisfies a relational task objective.

The central difficulty in applying NDFs to scenarios with changing pairs of objects is to assign a set of *consistent* local coordinate frames to the *task-relevant parts* of the objects in the demonstrations, which may be both *unaligned* and *differently shaped*. We propose an optimization method that uses two NDFs (one per object) and a *single* 3D keypoint label in just *one* of the demonstrations, to assign a set of local coordinate frames that are consistently posed relative to the task-relevant parts of the objects. We then apply NDFs to localize the corresponding coordinate frames for unseen pairs of objects presented in arbitrary initial poses, and solve for the relative transformation between them that satisfies the desired relation. However, errors can accumulate when inferring a relative transformation based on a pair of coordinate frames that have been *independently* localized. To mitigate this effect, we also propose a learning approach that directly models the *joint* configuration of the pair of objects and helps refine the transformation for satisfying the relation.

We validate R-NDFs on three relational rearrangement tasks in both simulation and the real world. Our simulation results show that R-NDFs outperform a set of baseline approaches, and our proposed optimization and learning-based refinement schemes benefit overall task success. Finally, our real world results exhibit the effectiveness of R-NDFs on pairs of diverse real world objects in tabletop pick-and-place, and highlight the potential for applying our approach to multi-step tasks.

## 2   Background: Neural Descriptor Fields

A Neural Descriptor Field (NDF) [3] represents an object using a function $f$ that maps a 3D coordinate $\mathbf{x} \in \mathbb{R}^3$ and an object point cloud $\mathbf{P} \in \mathbb{R}^{3 \times N}$ to a spatial descriptor in $\mathbb{R}^d$:

$$f(\mathbf{x}|\mathbf{P}) : \mathbb{R}^3 \times \mathbb{R}^{3 \times N} \to \mathbb{R}^d. \tag{1}$$

The function $f$ is parameterized as a neural network constructed to be SE(3)-equivariant, such that if an object is subject to a rigid body transform $\mathbf{T} \in \mathrm{SE}(3)$ its spatial descriptors transform accordingly[*]:

$$f(\mathbf{x}|\mathbf{P}) \equiv f(\mathbf{T}\mathbf{x}|\mathbf{T}\mathbf{P}). \tag{2}$$

This enables NDFs to behave consistently for the same object, regardless of the underlying pose. NDFs are also trained to learn correspondence over objects in the same category, so that points near similar geometric features of different instances (e.g., a point near the handle of two different mugs) are mapped to similar descriptor values. The equivariance property is obtained by using SO(3)-equivariant neural network layers [4] and mean-centered point clouds, while the category-level correspondence is obtained by training $f$ on a category-level 3D reconstruction task [3, 5].

---

[*]We use homogeneous coordinates for ease of notation, i.e., $\mathbf{T}\mathbf{x}$ denotes $\mathbf{R}\mathbf{x}+\mathbf{t}$ where $\mathbf{T} = (\mathbf{R}, \mathbf{t}) \in \mathrm{SE}(3)$.

NDFs can also be redefined to model a field over full SE(3) poses, rather than individual points. This is achieved by concatenating the descriptors of the individual points in a *rigid set* of query points $\mathcal{X} \in \mathbb{R}^{3 \times N_q}$, i.e., a set of three or more non-collinear points $\mathbf{x}_i$, $i = 1...N_q$, that are constrained to transform together rigidly. This construction allows NDFs to represent an SE(3) pose $\mathbf{T}$ via its action on $\mathcal{X}$, i.e., via the coordinates of the *transformed query point cloud* $\mathbf{T}\mathcal{X}$:

$$\mathcal{Z} = F(\mathbf{T}|\mathbf{P}) = \bigoplus_{\mathbf{x}_i \in \mathcal{X}} f(\mathbf{T}\mathbf{x}_i|\mathbf{P}) \tag{3}$$

Thus, $F$ maps a point cloud $\mathbf{P}$ and an SE(3) pose $\mathbf{T}$ to a category-level pose descriptor $\mathcal{Z} \in \mathbb{R}^{d \times N_q}$, where $F$ inherits the same SE(3)-equivariance from $f$.

## 3  General Problem Setup and Preliminaries

Our high level goal is to enable a user to specify a task involving a geometric relationship between a pair of rigid objects, and enable a robot to perform this task on unseen object instances presented in arbitrary initial poses. Examples of relations we consider include "*mug hanging on a rack*", "*bowl stacked upright on a mug*", and "*bottle placed upright on a tray*".

Concretely, our goal is to build a system that takes as input two (nearly complete) 3D point clouds $\mathbf{P}_A$ and $\mathbf{P}_B$ (each segmented out from the overall scene) of objects $\mathbf{O}_A$ and $\mathbf{O}_B$, and outputs an SE(3) transformation $\mathbf{T}_B$ for transforming $\mathbf{O}_B$ into a configuration that satisfies a desired relation between $\mathbf{O}_A$ and $\mathbf{O}_B$. We represent the relation as an alignment between a pair of local coordinate frames attached to task-relevant geometric features of the objects, and break down the problem of obtaining $\mathbf{T}_B$ into (i) assigning a set of consistent coordinate frames to the task-relevant local object parts and (ii) localizing these coordinate frames on the relevant parts of the new objects.

Furthermore, we assume a user specifies the relational task by providing a small handful of $K$ task demonstrations $\{\mathcal{D}_i\}_{i=1}^K$, such that it's intuitive and efficient to specify a wide diversity of tasks with minimal engineering effort. A demonstration $\mathcal{D}$ consists of point clouds $\hat{\mathbf{P}}_A$ and $\hat{\mathbf{P}}_B$ (of objects $\hat{\mathbf{O}}_A$ and $\hat{\mathbf{O}}_B$) and relation-satisfying transformation $\hat{\mathbf{T}}_B$.

**NDFs for Encoding Single Unknown Object Relations**. Prior work on NDFs may be applied to a simplified version of this task, where the geometry and state of $\mathbf{O}_A$ is *known*. Given that $\mathbf{O}_A$ is known, we can initialize a set of query points $\mathcal{X}_A$ near the task-relevant part of $\mathbf{O}_A$ and use the query points to encode the relative pose $\hat{\mathbf{T}}_B$ via Equation (3). Thus, a demonstration $\mathcal{D}$ is mapped to a target pose descriptor $\hat{\mathcal{Z}} = F(\hat{\mathbf{T}}_B^{-1}|\hat{\mathbf{P}}_B)$ representing the (inverse of the) final pose of $\hat{\mathbf{O}}_B$ *relative to* $\mathbf{O}_A$. In practice, pose descriptors from multiple demonstrations $\{\mathcal{D}_i\}_{i=1}^K$ are averaged to obtain an overall descriptor $\hat{\mathcal{Z}} = \frac{1}{K} \sum_{i=1}^K \hat{\mathcal{Z}}_i$ for the whole set, which has important implications in the version of the task with two unknown objects (see Section 4.1 for further discussion).

Given a novel object instance represented by point cloud $\mathbf{P}_B$, we can compute a transformation $\mathbf{T}_B$ such that transforming $\mathbf{O}_B$ by $\mathbf{T}_B$ satisfies the demonstrated relation between $\mathbf{O}_A$ and $\mathbf{O}_B$. This is achieved by minimizing the L1 distance to the target pose descriptor $\hat{\mathcal{Z}}$:

$$\mathbf{T}_B^{-1} = \underset{\mathbf{T}}{\operatorname{argmin}} \|F(\mathbf{T}|\mathbf{P}_B) - \hat{\mathcal{Z}}\|. \tag{4}$$

Intuitively, Equation (4) performs well across different objects due to the fact that NDFs are pretrained to enable reconstruction across a large dataset of 3D shapes. As a result, shared descriptors are discovered across different instances in a shape category. In contrast, training a model directly on the few demonstrations (e.g., for regressing pose $\mathbf{T}_B$) would be more susceptible to overfitting.

## 4  Method

We now describe how we apply NDFs to infer relations between pairs of unknown objects. In Section 4.1, we propose an iterative optimization method for assigning consistent task-relevant coordinate frames to multiple objects. In Section 4.2, we discuss how we train a neural network on top of NDF features to model the joint object configuration and refine an inferred transformation. The system inputs consist of pretrained NDFs $f_A$ and $f_B$ for each object category, demonstrations $\{\mathcal{D}_i\}_{i=1}^K = \{((\hat{\mathbf{P}}_A, \hat{\mathbf{P}}_B, \hat{\mathbf{T}}_B)_i)\}_{i=1}^K$, and a *single* labeled 3D coordinate $\mathbf{x}_{AB}$ for *one* of the demonstrations, indicating approximately where the respective demonstration objects interact.

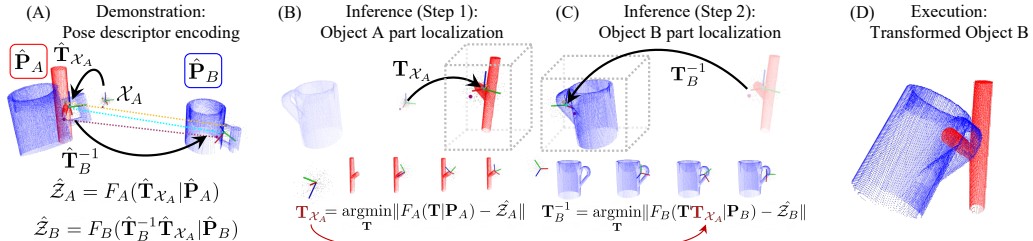

Figure 2: **Method Overview.** **(A)** A demonstration $(\hat{\mathbf{P}}_A, \hat{\mathbf{P}}_B, \hat{\mathbf{T}}_B)$ of a relation is encoded into a pair of pose descriptors by randomly sampling a set of query points $\mathcal{X}_A$ at the origin and transforming it by $\hat{\mathbf{T}}_{\mathcal{X}_A}$ to be near the task-relevant interaction point $\mathbf{x}_{AB}$. NDFs $f_A$ and $f_B$ are then used to obtain descriptors $\hat{\mathcal{Z}}_A$ and $\hat{\mathcal{Z}}_B$ representing coordinate frames near the task-relevant local parts on the objects. **(B)** Given point cloud $\mathbf{P}_A$ of a novel object, NDF $f_A$, and pose descriptor $\hat{\mathcal{Z}}_A$, pose $\mathbf{T}_{\mathcal{X}_A}$ of the corresponding coordinate frame on $\mathbf{P}_A$ is found. **(C)**. This procedure is then repeated with $\mathbf{P}_B$, $f_B$, and $\hat{\mathcal{Z}}_B$ to find pose $\mathbf{T}_B^{-1}$ of the relevant parts of $\mathbf{P}_B$, *relative to pose $\mathbf{T}_{\mathcal{X}_A}$ found in the first inference step*. **(D)** Transforming $\mathbf{P}_B$ by $\mathbf{T}_B$ satisfies the desired relation.

## 4.1 Multiple NDFs for Inferring Pairs of Task-Relevant Local Coordinate Frames

Consider a scenario where $\mathbf{O}_A$ and $\mathbf{O}_B$ have *unknown* underlying shapes and configurations. We now show how NDFs can be used for inferring a *pair* of task-relevant local coordinate frames on both objects and recovering a transformation $\mathbf{T}_B$ that satisfies the relation. The key idea of our approach is to formulate this problem as a bi-level optimization (illustrated in Figure 2), where we first optimize to find a task-relevant portion of $\mathbf{O}_A$, and subsequently optimize a relative transform of a local part of $\mathbf{O}_B$ with respect to the local region of $\mathbf{O}_A$.

We begin with two pretrained NDFs, $f_A$ and $f_B$, and query points $\mathcal{X}_A$ in a canonical pose at the world frame origin. We obtain $\mathcal{X}_A$ by sampling $N_q$ points from a zero-mean Gaussian and scaling such that $\mathcal{X}_A$ has scale similar to the salient object parts. We then use the keypoint $\mathbf{x}_{AB}$ to transform $\mathcal{X}_A$ near the task-relevant features in the demonstration associated with $\mathbf{x}_{AB}$. Denote this transformation as $\hat{\mathbf{T}}_{\mathcal{X}_A}$. Finally, we encode *world-frame* pose $\hat{\mathbf{T}}_{\mathcal{X}_A}$ into a descriptor conditioned on $\hat{\mathbf{P}}_A$, as $\hat{\mathcal{Z}}_A = F_A(\hat{\mathbf{T}}_{\mathcal{X}_A}|\hat{\mathbf{P}}_A)$, and *relative* pose $\hat{\mathbf{T}}_B^{-1}$ as $\hat{\mathcal{Z}}_B = F_B(\hat{\mathbf{T}}_B^{-1}\hat{\mathbf{T}}_{\mathcal{X}_A}|\hat{\mathbf{P}}_B)$, conditioned on $\hat{\mathbf{P}}_B$. At test-time, we optimize both the world-frame pose of the query points $\mathbf{T}_{\mathcal{X}_A}$ and the (inverse of) pose $\mathbf{T}_B$ relative to the initial pose found in the first step:

$$\mathbf{T}_{\mathcal{X}_A} = \underset{\mathbf{T}}{\operatorname{argmin}} \|F_A(\mathbf{T}|\mathbf{P}_A) - \hat{\mathcal{Z}}_A\| \quad (5) \qquad \mathbf{T}_B^{-1} = \underset{\mathbf{T}}{\operatorname{argmin}} \|F_B(\mathbf{T}\mathbf{T}_{\mathcal{X}_A}|\mathbf{P}_B) - \hat{\mathcal{Z}}_B\| \quad (6)$$

Figure 2 shows an example of this pipeline, where the resulting $\mathbf{T}_B$ is applied to the point cloud $\mathbf{P}_B$ of $\mathbf{O}_B$ to satisfy the "hanging" relation.

**Minimizing Descriptor Variance**. In practice, solving Equations (5) and (6) works better if pose descriptors $\{\hat{\mathcal{Z}}_i\}_{i=1}^K$ from multiple demonstrations are averaged together to obtain an overall target descriptor $\hat{\mathcal{Z}} = \frac{1}{K}\sum_{i=1}^K \hat{\mathcal{Z}}_i$ (see Sec. 6.1 and [3]). The reason is that a single demonstration *underspecifies* which object parts are relevant for the task, allowing $\hat{\mathcal{Z}}$ to be sensitive to object features which are not relevant to the desired relation. Instead, a set of demonstrations using slightly different objects (e.g., with different scales) reveals regions near local interactions that are *shared* across the demonstrations, which helps disambiguate between parts that are critical vs. irrelevant for the specified relation.

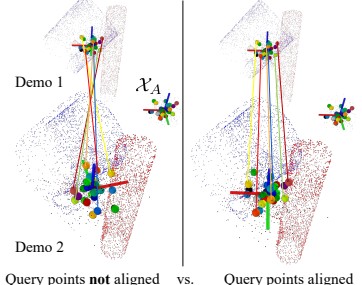

Figure 3: **Demo alignment.** We align the query points by minimizing the variance across the descriptor set before averaging.

However, to avoid the pitfalls of averaging across a potentially multimodal or disjoint set, we want descriptors in the set $\{\hat{\mathcal{Z}}_i\}_{i=1}^K$ to be sensitive to nearby local geometry in a way that is *consistent* (i.e., *unimodal*) across the demos. This only occurs if the *query points* used to obtain the descriptors are themselves consistently aligned relative to each respective object (see Figure 3). Therefore, we need

to find a transformation $\hat{\mathbf{T}}_{\mathcal{X}_A,i}$ for *each* demonstration $\mathcal{D}_i$ that transforms the canonical query points $\mathcal{X}_A$ into a configuration that leads the descriptors $\{\hat{\mathcal{Z}}_i = F_A(\hat{\mathbf{T}}_{\mathcal{X}_A,i}|\hat{\mathbf{P}}_{A,i})\}_{i=1}^K$ to be consistent with each other. We address this by finding the set of transformations $\{\hat{\mathbf{T}}_{\mathcal{X}_A,i}\}_{i=1}^K$ that minimizes the *variance* across the descriptor set $\{\hat{\mathcal{Z}}_i = F_A(\hat{\mathbf{T}}_{\mathcal{X}_A,i}|\hat{\mathbf{P}}_{A,i})\}_{i=1}^K$:

$$\min_{\{\hat{\mathbf{T}}_{\mathcal{X}_A,i}\}_{i=1}^K} \text{Var}(\{\hat{\mathcal{Z}}_i\}_{i=1}^K) \quad \text{subject to} \quad \hat{\mathcal{Z}}_i = F_A(\hat{\mathbf{T}}_{\mathcal{X}_A,i}|\hat{\mathbf{P}}_{A,i}) \quad \text{for } i = 1, ..., K \tag{7}$$

where $\text{Var}(\cdot)$ denotes the sum of the per-element variance across a set of vectors. We perform this minimization by applying NDFs in an alternating optimization procedure. Starting with an initial reference pose (constructed using $\mathbf{x}_{AB}$) placing $\mathcal{X}_A$ near the task-relevant object parts in one of the demonstrations, we iteratively apply Equation (5) to obtain a descriptor for each demonstration that matches the reference. At the outer level, we refit the reference descriptor using the mean of the most recently obtained individual descriptors, and repeat. More details can be found in the Appendix.

## 4.2 Capturing Joint Descriptor Alignment through Learned Energy Functions

The method in Section 4.1 proposes to infer a desired relation by *sequentially* localizing *independent* coordinate frames for each object. While this approach is generally effective, small errors can accumulate and cause slight misalignments that lead to failure in the execution. We thus propose to learn a neural network which directly captures the *joint* configuration of $\mathbf{O}_A$ and $\mathbf{O}_B$ that satisfies the desired relation, and use this model to refine predictions made by the method in Section 4.1.

**Pairwise Energy Functions**. We train an Energy-Based Model (EBM) $E_\theta(\cdot)$ [6] to parameterize a learned energy landscape over NDF encodings of relative poses between $\mathbf{O}_A$ and $\mathbf{O}_B$ (i.e,. $E_\theta$ acts as a learned analogue for the L1 distance in Section 4.1). The energy function $E_\theta(\cdot)$ is trained so that the ground truth transform of $\mathbf{O}_B$ with respect to $\mathbf{O}_A$ is recovered given NDFs $f_A$ and $f_B$ (note that $f$ corresponds to descriptor evaluation at *single* coordinate $\mathbf{x}$ while $F$ is defined over *sets* of coordinates). Explicitly, our energy function is trained so that:

$$\mathbf{T}_B = \underset{\mathbf{T}}{\text{argmin}} \left[ E_\theta(f_B(\cdot|\mathbf{T}\mathbf{P}_B), f_A(\cdot|\mathbf{P}_A)) \right]. \tag{8}$$

Since each NDF is a continuous field, it is difficult to input them directly into our energy function $E_\theta(\cdot)$. We represent the energy function as the sum of the point-wise evaluation of each NDF on a set of different query points $\mathcal{X}_E$ sampled from transformed pointcloud $\mathbf{T}\mathbf{P}_B$.

$$E_\theta(f_B(\cdot|\mathbf{T}\mathbf{P}_B), f_A(\cdot|\mathbf{P}_A)) = \sum_{\mathbf{x} \in \mathcal{X}_E} E_\theta(f_B(\mathbf{x}|\mathbf{T}\mathbf{P}_B), f_A(\mathbf{x}|\mathbf{P}_A)) \tag{9}$$

At test-time, we use Equation (8) to refine the transformation obtained using Equations (5) and (6).

## 4.3 Learning

**NDF training**. We represent NDFs $f_A$ and $f_B$ as two neural networks with identical architecture and separate weights. Following [3], the architecture consists of a PointNet [7] point cloud encoder with SO(3) equivariant Vector Neuron [4] layers, and a multi-layer perceptron (MLP) decoder. The NDF is represented as a function mapping a 3D coordinate and a point cloud to the vector of concatenated activations of the MLP. The models are trained end-to-end to reconstruct 3D shapes given object point clouds. We use a dataset of ground truth 3D shapes and generate a corresponding set of 3D point clouds in simulation. More architecture and training data details can be found in the Appendix.

**Energy-Based Model Training**. We supervise the EBM $E_\theta$ so that optimization over the learned energy landscape recovers the relative transform between $\mathbf{O}_A$ and $\mathbf{O}_B$. In particular, we follow the training objective in [8] and train $\arg\min_{\mathbf{T}}[E_\theta(\cdot)]$ to match a target pose using the following procedure. We first apply a small delta perturbation $\mathbf{T}_\Delta$ to $\hat{\mathbf{T}}_B\hat{\mathbf{P}}_B$ (i.e., the point cloud of $\mathbf{O}_B$ in its final configuration) to obtain $\hat{\mathbf{P}}_{B,\Delta} = \mathbf{T}_\Delta\hat{\mathbf{T}}_B\hat{\mathbf{P}}_B$. We then train $E_\theta$ to iteratively refine an initial random pose $\mathbf{T}_0$ with translation $\mathbf{t}_0$ and rotation $\mathbf{R}_0$ to *undo* the perturbation pose $\mathbf{T}_\Delta$. We run $n$ steps of optimization on $\mathbf{t}_0$ and $\mathbf{R}_0$, where an individual step is given by $\mathbf{t}_k = \mathbf{t}_{k-1} - \lambda\nabla_{\mathbf{t}}E_\theta(f_A(\cdot|\hat{\mathbf{P}}_A), f_B(\cdot|\mathbf{T}\hat{\mathbf{P}}_{B,\Delta}))$ and $\mathbf{R}_k = \mathbf{R}_{k-1} - \lambda\nabla_{\mathbf{R}}E_\theta(f_c(\cdot|\hat{\mathbf{P}}_A), f_B(\cdot|\mathbf{T}\hat{\mathbf{P}}_{B,\Delta}))$.

We may train the energy function so that $\mathbf{T}_n$ corresponds to the inverse of the perturbation pose $\mathbf{T}_\Delta$ using $\mathcal{L}_{\text{trans}} = \|\mathbf{t}_n - \mathbf{t}_\Delta^{-1}\|$ and $\mathcal{L}_{\text{rot}} = \|\mathbf{R}_n - \mathbf{R}_\Delta^{-1}\|$. However, with symmetric objects, there are multiple different rotations $\mathbf{R}_n$ which may satisfy the desired relation (e.g., a bowl is still "on" a

mug, regardless of the angle about its radial axis). To account for these symmetries, we implicitly enforce consistency between an optimized transform $\mathbf{T}_n$ and $\mathbf{T}_\Delta^{-1}$ by enforcing that its application on $\hat{\mathbf{P}}_{B,\Delta}$ leads to a similar point cloud to $\hat{\mathbf{T}}_B\hat{\mathbf{P}}_B$. We achieve this by minimizing the Chamfer loss [9] between the optimized transformed point cloud $\mathbf{T}_n\hat{\mathbf{P}}_{B,\Delta}$ and the demonstration point cloud $\hat{\mathbf{T}}_B\hat{\mathbf{P}}_B$.

## 5  Application to Tabletop Manipulation

**Robot and Environment Setup**. We apply the method in Section 4 to the problem of tabletop object rearrangement using a Franka Panda robotic arm with a Robotiq 2F140 parallel jaw gripper. The arm is used to collect the demonstrations and to execute the inferred transformation at test-time. Our environment consists of the arm on a table with four calibrated depth cameras.

**Providing and Encoding Demonstrations**. When collecting a demonstration, initial object point clouds $\hat{\mathbf{P}}_A$ and $\hat{\mathbf{P}}_B$ of objects $\hat{\mathbf{O}}_A$ and $\hat{\mathbf{O}}_B$ are obtained by fusing a set of back projected depth images. The demonstrator moves the gripper to a pose $\hat{\mathbf{T}}_\text{grasp}$, grasps $\hat{\mathbf{O}}_B$, and finally moves the gripper to a pose $\hat{\mathbf{T}}_\text{place}$ that satisfies the desired relation between $\hat{\mathbf{O}}_A$ and $\hat{\mathbf{O}}_B$. $\hat{\mathbf{T}}_B$ is obtained as $\hat{\mathbf{T}}_\text{place}\hat{\mathbf{T}}_\text{grasp}^{-1}$. In *one* of the demonstrations, a 3D keypoint $\mathbf{x}_{AB}$ is labeled near the parts of the objects that interact with each other by moving the gripper to this region and recording its position.

**Test-time Task Setup and Inference**. At test time, we are given point clouds $\mathbf{P}_A$ and $\mathbf{P}_B$ of new objects $\mathbf{O}_A$ and $\mathbf{O}_B$. Equations (5), (6), and (8) are applied in sequence to obtain $\mathbf{T}_B$. $\mathbf{T}_B$ is applied to $\mathbf{O}_B$ by transforming an initial grasp pose $\mathbf{T}_\text{grasp}$ (obtained using a separate grasp generation pipeline) by $\mathbf{T}_B$ to obtain a placing pose $\mathbf{T}_\text{place} = \mathbf{T}_B\mathbf{T}_\text{grasp}$, and off the shelf inverse kinematics and motion planning is used to reach $\mathbf{T}_\text{grasp}$ and $\mathbf{T}_\text{place}$.

## 6  Experiments and Results

Our experiments are designed to evaluate R-NDFs in executing relational rearrangement tasks with unseen objects using only a few demonstrations. We seek to answer three questions: (1) How well do R-NDFs predict transformations that satisfy a relational task? (2) How important is each component in R-NDFs? (3) Can R-NDFs be used to perform multi-object pick-and-place tasks in the real world?

**Baselines**. As existing rearrangement methods are not directly applicable with so few demonstrations, we compare with two constructed baselines. The first is to train an MLP to directly regress the relative transformation between objects ("Pose Regression"). The MLP takes as input the point cloud encodings obtained from the same PointNet [7] encoder with Vector Neuron [4] layers used in NDFs, and is trained directly on the demonstrations. The second method is based on 3D point cloud registration ("Patch Match"). We use a state of the art registration method [10] to align the test-time shapes to the demonstration shapes and then compute the resulting relative transformation.

**Task Setup and Evaluation Metrics**. We consider three relational rearrangement tasks for evaluation: (1) Hanging a mug on the hook of a rack, (2) Stacking a bowl upright on top of a mug, and (3) Placing a bottle upright inside of a box-shaped container. We provide 10 demonstrations of each task and evaluate if each method, using the demonstrations, can infer a transformation that satisfies the desired relation for unseen pairs of object instances with randomly sampled poses. Experiments are conducted in both the real-world and in simulation using PyBullet [11]. In simulation, the transformation obtained by each method is directly applied by resetting the simulator to the transformed object states. To quantify performance, we report the success rate over 100 trials, where we use the ground truth simulator state to compute success (objects must be in contact, have the correct relative orientation, and not interpenetrate).

### 6.1  Simulation Results

We begin by evaluating how well R-NDFs can infer the desired transformations in simulation. We consider two settings varying difficulty. First, the pair of unseen objects are positioned randomly on the table with a randomly sampled "upright" orientation (similar to those used in the demonstrations). Second, the orientation of $\mathbf{O}_B$ is randomly sampled from the full space of 3D rotations.

Results in Table 4a compare the performance of our approach to the baselines. We find that training a model to directly regress the relative pose using the point cloud embeddings leads to substantial

| Method | Bowl on Mug | | Mug on Rack | | Bottle in Container | |
|---|---|---|---|---|---|---|
| | Upright | Arbitrary | Upright | Arbitrary | Upright | Arbitrary |
| Pose Regression | 35.0 | 6.0 | 13.0 | 10.0 | 37.9 | 12.0 |
| Patch Match | 34.0 | 32.0 | 56.0 | 44.0 | 44.0 | 42.0 |
| R-NDF | **74.0** | **70.0** | **84.0** | **75.0** | **80.0** | **75.0** |

(a)

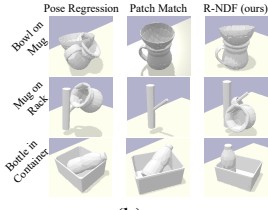

(b)

Figure 4: (a) **Relation inference success rates in simulation.** R-NDF performs better than the baseline approaches. (b) **Example predictions.** Representative predictions made by each method in simulation

| Multiple Demonstration | Query Point Alignment | EBM Refinement | Upright Pose | Arbitrary Pose |
|---|---|---|---|---|
| No | No | No | 39.3 | 43.6 |
| Yes | No | No | 66.0 | 60.0 |
| Yes | Yes | No | 78.0 | 72.0 |
| Yes | Yes | Yes | 84.0 | 75.0 |

(a)

(b)

Figure 5: (a) **Ablations.** R-NDF performance with different components ablated. Success rate is highest when using multiple demonstrations, query point alignment, and EBM refinement. (b) **Success vs. Keypoint Noise.** Success rate vs. magnitude of noise (normalized by object size) added to the single labeled 3D keypoint $\mathbf{x}_{AB}$.

overfitting and much lower success rates. On the other hand, the registration-based method can sometimes find transformations that correctly align the unseen shapes to the demonstration objects, and thus achieves higher success rates than pose regression. However, 3D registration is susceptible to locally optimal results that align the task *irrelevant* parts of the objects. Common failure modes of using 3D registration in the tasks we consider include aligning the body of the mug but ignoring the handle, or aligning the racks to be upside down. Figure 4b illustrates the final simulator state after applying some of the representative predictions of each method.

In contrast, R-NDFs more accurately localize the task-relevant object parts and assign coordinate frames to these parts that are consistent with the demonstrations, leading to the highest success rates. Consistent with [3], the performance gap between the "upright" and "arbitrary" pose settings is small, which can be attributed to the built-in equivariance of the features used in R-NDF.

## 6.2 Ablations

Next, we analyze the importance of the individual components of R-NDFs. We investigate ablations on the simulated "mug on rack" task, again considering both "upright" and "arbitrary" pose settings.

The top row of Table 5a illustrates that R-NDF performs worse with a single demonstration. Since there are multiple possible explanations for the alignment between two objects when given one example of the desired relation, pose descriptors obtained from a single demonstration are more sensitive to task irrelevant object features. The second row of Table 5a investigates the effect of averaging descriptors across the set of demonstrations without first aligning the query points relative to the objects in each demo. We modified the demonstrations to provide keypoints $\{\mathbf{x}_{AB,i}\}_{i=1}^{K}$ near the relevant region in *each* demonstration, and then transform the query points to this region *without* aligning their orientations. Removing the query point alignment reduces the performance. The third row of Table 5a shows that removing the EBM refinement also decreases the success rate.

We further examine the importance of accurately specifying the 3D keypoint $\mathbf{x}_{AB}$ near the task-relevant region on one of the demonstrations. We run the trials multiple times with Gaussian distributed noise added to the labeled point. Figure 5b shows a plot of the success rate vs. the noise magnitude normalized by the approximate size of the object. The plot indicates that with limited noise perturbation, the success rate does not suffer significantly, though we observe a steep decline with more substantial perturbations. These larger perturbations shift the query points to regions near geometric features that are less relevant to the desired relation.

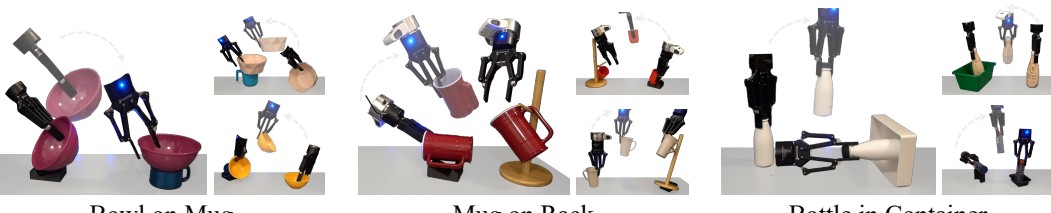

| Bowl on Mug | Mug on Rack | Bottle in Container |

Figure 6: **Real Execution Results.** Example executions of relational tasks on unseen mugs, bowls, bottles, racks, and containers in the real world. Our framework enables inferring the relative transformation between pairs of unseen objects in arbitrary initial poses from a small handful of unaligned demonstrations of each task.

### 6.3 Real Results

Finally, we validate that R-NDFs can be used to perform pick-and-place on pairs of unseen objects in the real world. Figure 6 shows the execution on our three tasks. Our method successfully infers a tranformation between the objects that satisfies the relations, despite the objects being presented in a challenging array of initial configurations. Figure 1 shows a multi-step rearrangement application of R-NDFs for the "bowl on mug" task. First, a relation between the *mug* and the *table* is specified and inferred for placing the mug upright. Then, the system executes the "stacking" relation between the *bowl* and the *upright mug*. This highlights how R-NDFs can enable executing sequential chains of relations to satisfy task objectives involving more than two objects. Please see our attached supplemental video for additional real world results.

## 7 Related Work

**Novel Object Rearrangement**. Several methods exist for novel object rearrangement [1, 12–29], many of which don't consider multiple varying objects that interact. CatBC [30] uses dense correspondence models to achieve impressive pick-and-place policy generalization from a single demonstration, but assumes a known receptacle for placing. Neural shape mating [31], OmniHang [32], and kPAM 2.0 [2] generalize to pairs of unseen objects, but these approaches train on large task-specific datasets. TransporterNets [33, 34] enables rearrangement with varying pick and place locations from a few demonstrations, but focuses on top-down manipulation and struggles with out-of-plane reorientation. In contrast, we focus on executing relations involving large 3D reorientations.

**Neural Fields in Robotics**. Neural fields use neural networks to parameterize functions over continuous spatial or temporal coordinates [35]. They have been applied to model various signals and scene properties, such as images [36], geometry [5, 37, 38], appearance [39, 40], tactile imprints [41], and sound [42], with high fidelity and memory efficiency. Neural fields have been applied to represent objects for manipulation [3, 43–46] and environment states for dynamics and policy learning [47–49]. They have also been used for pose estimation [50, 51], SLAM [52, 53], and representing object geometry without depth cameras [54, 55].

## 8 Limitations and Conclusion

**Limitations**. R-NDFs require a pretrained NDF for each category used in the task, which can be nontrivial to obtain for novel object categories without existing 3D model datasets. Our approach also requires an annotated keypoint to localize task-relevant object parts. Future work could explore automated discovery of task-relevant regions directly from a set of demonstrations. Our system uses depth cameras, which often struggle with noise and objects with thin and transparent features. An RGB-only approach offering a similar level of generalization would be interesting to investigate. Finally, we require segmented object point clouds. While object instance segmentation is quite mature, pretrained segmentation models regularly struggle when objects are in diverse orientations.

**Conclusion**. This work presents an approach for learning from a limited number of demonstrations to rearrange novel objects into configurations satisfying a relational task objective. We develop methods that build upon prior applications of neural fields for representing objects and increase the scope of tasks they can achieve. Our results illustrate the general applicability of our framework across a diverse range of relational tasks involving pairs of novel objects in arbitrary initial poses.

## Acknowledgments

This work is supported by the NSF Institute for AI and Fundamental Interactions, DARPA Machine Common Sense, NSF grant 2214177, AFOSR grant FA9550-22-1-0249, ONR grant N00014-22-1-2740, MIT-IBM Watson Lab, MIT Quest for Intelligence and Sony. Anthony Simeonov and Yilun Du are supported in part by NSF Graduate Research Fellowships. We thank the members of the Improbable AI Lab and the Learning and Intelligent Systems Lab for the helpful discussions and feedback on the paper.

## Author Contributions

**Anthony Simeonov** developed the idea of minimizing descriptor variance for aligning multiple demonstrations, set up the simulation and real robot experiments, played a primary role in paper writing, and led the project.

**Yilun Du** came up with and implemented the energy-based modeling framework for relative pose inference, helped develop the overall framework of using NDFs for relational rearrangement tasks, ran simulated experiments, helped with writing the paper, and co-led the project.

**Yen-Chen Lin** participated in research discussions about different ways to approach 6-DoF pick-and-place/rearrangement tasks, helped suggest improvements to the NDF training and optimization procedure, and helped with editing the paper.

**Alberto Rodriguez** helped with early brainstorming on how multiple NDF models could be used for multi-object rearrangement tasks and gave feedback on the tasks and real robot results.

**Leslie Kaelbling** helped develop the idea of chaining multiple pairwise relations together to perform multi-step tasks, provided suggestions on interesting rearrangement tasks to solve, and helped write and edit the paper.

**Tomás Lozano-Peréz** also helped suggest the application to multi-step tasks via sequencing relations, reinforced the investigation of representations grounded in local interactions between object parts, and provided valuable feedback on the paper.

**Pulkit Agrawal** was involved in early technical discussions about how to use multiple NDF models for rearrangement tasks, helped clarify key technical insights regarding query point labeling in the demonstrations, advised the overall project, and helped with paper writing and editing.

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
