# OpenReview forum: "SE(3)-Equivariant Relational Rearrangement with Neural Descriptor Fields"
_robot-learning.org/CoRL/2022/Conference — CoRL 2022 Poster_

### Official Review · Reviewer_krtE · 2022-07-06

**Originality:** Good
**Technical Quality:** Good
**Clarity Of Presentation:** Good
**Impact:** 3

**Recommendation:**

Weak Accept: I recommend accepting the paper, but will not argue for my recommendation if the majority of other reviewers have a different opinion.

**Summary:**

The paper builds on Neural Descriptor Fields and extends it to be used on two arbitrary objects (as opposed to one unknown and one fixed).
This is a more difficult problem as the query pointcloud location has to be defined from demonstrations.
The paper proposes an optimization method to consistently arrive at initial pointcloud location with a help of annotating its (initial) center.
After the pose of the query is defined on object A using NDF of A, its instance on object B is also defined using NDF of B, yielding pose descriptors Z_A and Z_B.
These pose descriptors are averaged over demonstrations.
During inference first the query pointcloud pose is defined and then the relative object pose is computed via optimization.
To improve accuracy further the paper also proposes the minimization of an Energy Based Model that fine-tunes the final pose.
The method was tested in simulations and real experiments.

**Issues:**

Please address the limitations of the full training and inference pipeline.

Please comment on scaling to more objects, e.g., what happens with 3 objects? For example, place bottle left to the plate and right to the box.

Can NDFs be used for tasks other than pick-and-place?


**Quality Of The Limitations Section:**

Additional details required

**Reviewer Expertise:**

5: The reviewer is absolutely certain that the evaluation is correct and very familiar with the relevant literature

**Robotics Focus:**

Sufficient demonstration on hardware

**Strengths And Weaknesses:**


Strengths

The paper addresses a relevant topic in robot learning and manipulation.
NDF is a promising and novel method to encode relative object poses from pointcloud recordings and it shows good potential for generalization to similar objects.
This paper further improves its applicability to two-object-class scenarios, where the two object instances might have been unseen during training.
The paper presents a reasonable and technically mostly sound approach to achieve this goal and shows evaluations to support the desgin choices.
The clarity of the paper is also good, it is easy to follow.
The figures illustrate the approach well.

Weaknesses

I found the story around the Energy Based Model minimization vague.
While it seems to mitigate the problem of inaccurate poses (after solving (4) and (5)), for me it is not entirely clear what is the source of pose error.
The paper discusses the accumulation of errors, but why do these error appear in the first place.
Does it relate to inaccurate demonstrations, limited expressiveness of NDFs, quality of the pointcloud and its segmentation?

The overall solution consists of multiple steps starting from camera calibration, training NDFs and pose descriptors till inference in real time, point cloud segmentation, optimizations to derive relative pose and grasp pose, etc.
While a few failure points (e.g. quality of depth) have been mentioned in the limitation section, more discussion on the limitation of the whole algorithmic pipeline would be welcome.
It seems to take a substantial effort to implement the algorithm, what are the key components to pay attention to, what are prone to fail?


**Summary Of Recommendation:**

The paper provides a well motivated improvement to NDFs, the technical quality and clarity are good, but could use more improvements (see Strengths & Weaknesses). The simulation and real world experiment support the claim of the paper. More insights into further limitations and implementation details could improve my rating.

---

> ### Author Response · Authors · 2022-08-23
> **Response to Reviewer krtE**
>
> > EBM story is vague
> >
>
> The following explanation tries to clarify. For “mug on rack”, there is some tolerance for slight imprecision when localizing the handle (because of the hole size + the thickness of the handle) and in localizing the peg (because of the angle and the length), such that if one step was perfect and the other was “just a little off”, the execution could often still succeed. However, there are cases where *both* will be “a little off”, and since it’s “a little off” on top of “a little off”, the overall error grows and the execution has a higher chance of failing. The EBM removes one step of estimation and refines the solution by looking at both objects jointly, as if their joint configuration were a single variable to directly optimize.
>
> > What’s the source of error? why do these errors appear in the first place? Does it relate to inaccurate demonstrations, limited expressiveness of NDFs, quality of the pointcloud and its segmentation?
> >
>
> Thanks for pointing this out. In general, there can be multiple sources of error in NDF descriptor matching. We have included an expanded discussion on these potential failure modes in Appendix A11.2.8. We also included a compact list below, roughly in order of importance:
> - Out-of-distribution/severe point cloud noise
> - Inaccuracies in the demonstrations
> - NDFs that pick up on task-irrelevant features
>
> > limitation of the whole algorithmic pipeline would be welcome. It seems to take a substantial effort to implement the algorithm, what are the key components to pay attention to, what are prone to fail? — More insights into further limitations and implementation details could improve my rating. … Please address the limitations of the full training and inference pipeline.
> >
>
> This is a fair point, R-NDF would perhaps constitute one part of a larger system that can perform general-purpose multi-object rearrangement. We have added a detailed list of what we believe to be the most important factors in Appendix A11.2.8, along with a compact list below:
> - Pretraining NDF on diverse point clouds (different shapes, occlusions, scalings, etc.)
> - Ensuring underlying 3D reconstruction doesn’t underfit
> - Use a few demonstrations with different shapes (more than one demo, and ensure demos aren't too similar)
> - Accurate point cloud segmentation and outlier removal
> - Run NDF optimization from several initializations
> - Collision-free planning with waypoints that stay away from potential collision
>
> > Please comment on scaling to more objects, e.g., what happens with 3 objects? For example, place bottle left to the plate and right to the box.
> >
>
> Great question, R-NDF can be extended to handle more than two objects. One way is by matching descriptors for more than one NDF at a time, another is by sequentially solving for relative transformations and composing them all together. We included an example of applying this idea to the task you described, where the goal is to place a mug upright next to a bottle and next to a bowl (see updated Appendix A8 for visualizations and discussion, along with the below paragraph for additional details).
>
> In general, there is evidence in the literature that optimization-based methods are compatible with composing multiple task costs/constraints. Recent prior work has set up a similar compositional approach for motion generation [1]. For kinematics, recent work has shown EBMs are useful for taking robot kinematics into account by setting up joint space decision variables and using a differentiable FK module [2]. This would allow adding additional joint space costs/constraints. In machine learning, energy-based models have been useful for compositional generative modeling [3-7].
>
> > Can NDFs be used for tasks other than pick-and-place?
> >
>
> NDFs can be used wherever it’s generally useful to localize a coordinate frame near a task-specific local region/part of an object/scene. Pick-and-place is one such setting, but there may be others. For example, matching fingertip poses in multi-finger grasping or solving for a geometric interaction between a tool and the environment. Currently, NDFs cannot be used for anything other than purely geometric tasks. See updated Appendix A11 for further discussion.
>
> [1] Urain, et al., “Composable Energy Policies for Reactive Motion Generation and Reinforcement Learning”, RSS 2021
>
> [2] Ganapathi, et al., “Implicit Kinematic Policies: Unifying Joint and Cartesian Action Spaces in End-to-End Robot Learning”, ICRA 2022
>
> [3] Du, et al., Compositional Visual Generation with Energy Based Models, NeurIPS 2020
>
> [4] Liu, Li, and Du, et al., Learning to Compose Visual Relations, NeurIPS 2021
>
> [5] Du, et al., Unsupervised Learning of Compositional Energy Concepts, NeurIPS 2021
>
> [6] Du, et al., Learning Iterative Reasoning through Energy Minimization, ICML 2022
>
> [7] Liu, Li, and Du, et al., Compositional Visual Generation with Composable Diffusion Models, ECCV 2022

---

### Official Review · Reviewer_haL1 · 2022-07-27

**Originality:** Very Good
**Technical Quality:** Good
**Clarity Of Presentation:** Fair
**Impact:** 3

**Recommendation:**

Weak Accept: I recommend accepting the paper, but will not argue for my recommendation if the majority of other reviewers have a different opinion.

**Summary:**

This paper presents R-NDF, a framework for pair-wise object rearrangement that is robust to pose variations in SE(3). The framework builds on prior work in Neural Descriptor Fields (NDFs) – an approach for learning spatial descriptors of objects that are robust to 6-DoF transformations. NDFs are limited to a single object, whereas R-NDF can learn geometric relationships between pairs of rigid-bodies. Experimental evaluations include simulated experiments and baseline comparisons to simple pose regression and patch matching. The results also include some ablations that investigate individual components of R-NDF. Qualitative results include real-robot demonstrations, which showcase R-NDF’s ability to precisely place mugs on wooden racks with just a handful number of demonstrations.


**Issues:**

See issues mentioned in the Weaknesses section.

**Quality Of The Limitations Section:**

Limitations are addressed clearly

**Reviewer Expertise:**

4: The reviewer is confident but not absolutely certain that the evaluation is correct

**Robotics Focus:**

Sufficient demonstration on hardware

**Strengths And Weaknesses:**

**Strengths**

+ On the whole, the idea of using SE(3) equivariant frameworks for robot perception is an interesting and practical direction for vision-based manipulation. Naive image-based perception systems struggle to generalize to unseen object poses and are not robust to scene changes. Equivariance is a property that has been crucial to data efficiency and robustness in 2D vision (like with UNets), so it’s great to see equivariance being adapted to 6-DoF manipulation in 3D settings, more specifically in multi-object scenarios. While data-augmentation is one possible alternative, it’s hard to guarantee robustness to pose-variations like with equivariant frameworks.

+ R-NDF is evaluated both in simulated and real-world setups. Particularly, the real-world results are compelling. The framework can imitate high-precision tasks with just 5-10 demonstrations, and do this with any given pose-variation. Also, this robustness to pose-variations is evaluated with simulated experiments, which might be easily reproducible if the authors open-source their code-base.

+ The figures in the paper are generally helpful. Figure 1 does a good job of providing an overview of the problem. Figure 2 provides a good summary of the approach and the high-level concepts. The experimental setup in Section 6 is mostly clear, and also includes a good set of ablations investigating individual aspects of the framework like energy-model based refinement and aligning query points.

**Weaknesses**

- One major concern is that R-NDF makes a lot of assumptions specific to the two-object setup, which could be better highlighted in the paper. Specifically, R-NDF assumes access to: (1) object segmentations, good object point-clouds, and approximate object localization (at least a centroid), (2) no distractor objects with only two-objects in the scene, (3) collision-free motion plans, and (4) pre-trained NDFs for each object.

(1) Object segmentation and localization might be obtainable for large rigid-body objects, but it’s unclear how robust R-NDF would be in cluttered scenes with heavy occlusions and inaccurate segmentations.

(2) The EBM formulation in Section 4 is also highly-engineered towards a setup with two-rigid objects. It’s unclear how easily extendible or generally applicable R-NDF would be to N-object rearrangement or anything that involves more than a single-step geometric transformation.

(3) Section 5 briefly mentions that a motion-planner is used to reach grasp and place poses, but how is the collision avoidance handled? Particularly for the “Mug on Rack” example, how does the motion-planner know not to collide with the wooden stem before hanging the handle on it?

(4) How long does it take to pre-train the NDFs? The paper claims only 5-10 demonstrations are needed to learn rearrangement transformations, but there is also the cost of pre-training NDFs on object instances, right?

These limitations don’t necessarily diminish the contributions of the paper, but it might be worth highlighting these requirements for future practitioners trying to adapt the framework to real-world manipulation challenges.

- Another issue is that the relationship between R-NDF and k-PAM (Gao et al, Manuelli et al) is not sufficiently discussed. The related work section states “.. kPAM 2.0 generalize to pairs of unseen objects, but these approaches are trained on large task-specific datasets” but the methods section for R-NDF states that “NDFs are pretrained to enable reconstruction across a large dataset of 3D shapes”. So does kPAM use more shapes than R-NDF or not? Also, is there anything preventing the experiments from using kPAM (Gao et al) as a baseline for the experiments?

- Minor: Section 4 could be re-written for better readability. There is a lot of linear algebra interweaved with the text, which makes the sentences hard to read at times. The text could be made more accessible by providing high-level intuitions before delving into mathematical definitions.

References: \
Gao et al: [https://groups.csail.mit.edu/robotics-center/public_papers/Gao20.pdf](https://groups.csail.mit.edu/robotics-center/public_papers/Gao20.pdf) \
Manueli et al: [https://arxiv.org/abs/1903.06684](https://arxiv.org/abs/1903.06684)


**Summary Of Recommendation:**

This paper proposes a novel SE(3) equivariant framework for object-rearrangement and achieves compelling real-world results with limited data. There are some issues with the presentation of methods and results, but the framework itself could be of interest to the manipulation community.

**Post Rebuttal**
The authors spent a lot of time and effort updating the paper and adding new sections, and I really appreciate that! These updates address most of my concerns.

My only reservation is still the strong assumption about "large rigid objects". As pointed out, there has been a lot of progress in instance segmentation and 3D encoders, but it's still unclear how R-NDFs would work with objects that, to begin with, are hard to represent with segmentations or pointclouds in the real-world like particles, clothes, and liquids. But nonetheless, "larger rigid objects" still cover a broad category of manipulation problems, and overall, this paper is quite relevant to the CoRL audience. So I am keeping my score of Weak Accept.

---

> ### Author Response · Authors · 2022-08-23
> **Response to Reviewer haL1**
>
> > Assume object segmentation and coarse localization. Potential difficulty with clutter, heavy occlusions, and noisy segmentation
> >
>
> Thanks for raising this important point. We included additional qualitative results highlighting that our approach can work with partial point clouds, but that it depends on the severity of the occlusion and exactly what parts are occluded (see updated Appendix A9).
>
> We note that significant progress has been made on instance segmentation [1,2] with other systems deploying these as components [3,4]. We thus believe it’s not unreasonable to assume access to segmented point clouds. But it’s true that these systems are not yet robust enough to be full “plug and play”, and severe occlusion/noise does cause an issue.
>
> Reducing the dependence on performant off-the-shelf perception modules also has the potential to expand the capabilities, but in a way that is roughly orthogonal to our proposed approach. Recent works on using different neural network encoders for 3D data that use local features have shown promising results in not requiring accurate segmentation [6]. The implications of global vs. local encoding on generalization and robustness have been discussed in various recent papers on neural fields [7,8]. Transferring such approaches to our setting ought to provide similar benefits.
>
> > EBM formulation engineered toward “two rigid object setup” … unclear how applicable R-NDF would be for N-object rearrangement
> >
>
> Great point. R-NDF is indeed applicable to rearrangement with more than two objects. Because we use optimization to solve for the pose, it’s straightforward to compose multiple pairwise cost terms. We provided a qualitative example of this for 3 objects where the task is to place a mug upright both “next to the bowl” and “next to the bottle”. We run the NDF optimization to recover a coordinate frame that satisfies both  “next to” relations simultaneously, by computing the optimization loss as the sum of the separate “bowl” and “bottle” descriptor matching losses. We also showed another example of 3-object rearrangement using the same “two relations inferred in sequence” used to obtain the result in Figure 1. See the results in updated Appendix A8 for more details.
>
> > Planning — how does the motion planner know not to collide with the wooden stem before hanging the handle on it?
> >
>
> Thanks for pointing out that we didn’t make this clear. While motion-planning constraints can be incorporated into R-NDF (see qualitative results in Appendix A7), this was not our main focus. We therefore simplified the motion planning problem by specifying an intermediate “waypoint” which the robot reaches between the pick-and-place to help avoid collisions. Appendix A11 discusses this in more detail.
>
> > How long does it take to pre-train the NDFs?
> >
>
> We train each NDF for around 150k iterations, which takes about half a day on one RTX 3090 GPU (some more details on the training are included in Appendix A1.3). While this is a nontrivial amount of time to wait, it only has to be done once per category using data obtained in simulation, after which the user can do perform new tasks using the same NDF model.
>
> > The relationship between R-NDF and k-PAM — Does kPAM use more shapes than R-NDF? Is there anything preventing the experiments from using kPAM as a baseline?
> >
>
> Thanks for the good question. In kPAM, they don’t necessarily have more shapes, but they have human-annotated keypoint labels on the full dataset. In our setting, this would be the equivalent of having demonstrations for every shape in the full offline dataset (used to pretrain the NDFs), whereas our method applies when we only have demos for a small handful of objects. Therefore, they should be considered an algorithm with access to privileged data, rather than a baseline. The analogous baseline version of their approach would be the pose regression comparison we performed, wherein we apply vanilla supervised learning applied on the small training dataset obtained from the demos.
>
> > Section 4 could be re-written
> >
>
> We have updated the writing in this section to try and make it easier to follow, thanks for the suggestion.
>
> [1] Xie, et al., “Unseen Object Instance Segmentation for Robotic Environments”, TRO 2021
>
> [2] He, et al., “Mask R-CNN”, ICCV 2017
>
> [3] Mousavian, et al., “6-DOF GraspNet: Variational Grasp Generation for Object Manipulation”, ICCV 2019
>
> [4] Danielczuk, et al., "Object Rearrangement Using Learned Implicit Collision Functions”, ICRA 2021
>
> [5] Sundermeyer, et al., “Contact-GraspNet: Efficient 6-DoF Grasp Generation in Cluttered Scenes”, ICRA 2021
>
> [6] Ryu, et al., “Equivariant Descriptor Fields: SE(3)-Equivariant Energy-Based Models for End-to-End Visual Robotic Manipulation Learning”, arXiv 2022
>
> [7] Xie, et al., “Neural Fields in Visual Computing and Beyond”, Computer Graphics Forum 2022
>
> [8] Peng, et al., “Convolutional Occupancy Networks”, ECCV 2020

---

> > ### Comment · Reviewer_haL1 · 2022-08-25
> > **Response to authors**
> >
> > Thank you for the detailed response, new write-ups, and figures. The authors spent a lot of time and effort updating the paper and adding new sections, and I really appreciate that! These updates address most of my concerns.
> >
> > My only reservation is still the strong assumption about "large rigid objects". As pointed out, there has been a lot of progress in instance segmentation and 3D encoders, but it's still unclear how R-NDFs would work with objects that, to begin with, are hard to represent with segmentations or pointclouds in the real-world like particles, clothes, and liquids. But nonetheless, "larger rigid objects" still cover a broad category of manipulation problems, and overall, this paper is quite relevant to the CoRL audience. So I am keeping my score of Weak Accept.

---

### Official Review · Reviewer_1zr8 · 2022-07-31

**Originality:** Fair
**Technical Quality:** Very Good
**Clarity Of Presentation:** Very Good
**Impact:** 3

**Recommendation:**

Weak Accept: I recommend accepting the paper, but will not argue for my recommendation if the majority of other reviewers have a different opinion.

**Summary:**

The paper proposes a way to infer the SE(3) transform to move an object (represented by a point cloud) to satisfy some required relation with another object (also represented by a point cloud) e.g. to hang a mug on a rack. Both objects should belong to some a priori known classes, for each of which a dataset of point cloud instances should exist. These datasets are used to pre-train neural networks that allow subsequent inference of a transform from a small number (around 10) demonstrations. The proposed approach is evaluated on a set of experiments, including a real robot setting, and compared with two baselines showing improvements.

**Issues:**

Please comment or act on the following.

- Line 228. Do you train the MLP on 10 examples only? If so, how do you manage to get something reasonable from it?
- Figure 6. The container lying on a side with a bottle about to drop when the manipulator lets go of it is a genuine situation from an experiment or an improperly rotated picture?
- In your bibliography, you have a number of arXiv references for papers that were published. For some entries you have page numbers and for some you do not, same for urls. Please make the bibliography consistent and cite the published versions of papers.


**Quality Of The Limitations Section:**

Limitations are addressed clearly

**Reviewer Expertise:**

3: The reviewer is fairly confident that the evaluation is correct

**Robotics Focus:**

Sufficient demonstration on hardware

**Strengths And Weaknesses:**

The paper builds heavily on the previous work on Neural Descriptor Fields. This work enabled inferring an SE(3) transformation for moving one object to satisfy a certain relation with the other when this other object is stationary and fixed. On the one hand, this paper goes in the right direction by removing this superfluous assumption. On the other hand, the path to this is rather incremental: two Neural Descriptor Fields are used instead of one. Additional refinement is added by using an additional multilayer perceptron resulting in a slight improvement in performance. Examples, including one with a real robot, that show improvements over some baselines are present. However, no code is provided in the supplementary and no promise of publishing the code upon acceptance is made which, in my opinion, is a big drawback.

**Minor Comments**

1. Equation (2). The property seems to be invariance rather than equivariance: usually equivariance of F:X->Y means that F(Tx)=TF(x) where T is an element of a group that acts on both X and Y, while invariance is F(Tx)=F(x). Equation (2) is similar to the latter.
2. Footnote on page 2. “T = (R, T)” probably should be “T = (R, t)”.
3. Line 93. “it’s” -> “it is”.
4. Line 106 and other places. I suggest using the usual “l^1” instead of “L1”.
5. Line 117. Maybe change “3D coordinate” to “3D point”. In my humble opinion a coordinate means each of the three numbers in a tuple representing a 3D point, not the whole tuple.
6. Equation (7). As far as I understand this is not a constrained optimization problem. Phrasing it as a constrained one seems misleading to me.
7. Lines 166-167. You write “small errors can accumulate”. What is the iterative procedure meant here? Is this the two-step procedure defined by Equations (5) and (6). If so, the word “accumulate” reads a bit weird for a two-step procedure.
8. Figure 5 (b). At the bottom left the “0.00” and “0.0” interpenetrate.

**Summary Of Recommendation:**

Although being somewhat incremental, the paper makes some progress in a relevant problem. It does not feature a (promise of) an open source implementation of the method, which is a big drawback.

---

> ### Author Response · Authors · 2022-08-23
> **Response to Reviewer 1zr8**
>
> > no code is provided in the supplementary and no promise of publishing the code upon acceptance is made
> >
>
> Thanks for pointing this out, we do plan to release code when we clean it up and our paper is made public. In the meantime, we have attached an anonymized zip file of our code. While the experiments may not be runnable in this current form, hopefully this acts as a sign that we will make good on publicly releasing our code in the future.
>
> > [EBM overfitting] Do you train the MLP on 10 examples only? If so, how do you manage to get something reasonable from it?
> >
>
> Please see our general rebuttal response.
>
> > [Sideways container] The container lying on a side with a bottle about to drop when the manipulator lets go of it is a genuine situation from an experiment or an improperly rotated picture?
> >
>
> This scenario was real and was set up to show the ability to handle varying poses of both object A and object B. We simply did not have the robot release the object for this run. Similar to the full 2-step example we show in the supplementary video for first executing “upright mug on the table” and then executing ”bowl on mug”, one could do the same in this case (i.e, start with “upright container on table” followed by ”bottle in container”). The solution would just be the rigid transformation applied to the container which makes it upright, also applied to this relative transformation of the bottle. We included a similar example in the new qualitative results (see updated Appendix A8), with a “mug on container, bottle in mug” task. Inferring the “bottle in mug” for a sideways mug results in a sideways bottle, but both shapes get transformed to be upright after determining how to execute the “mug in container” component.
>
> > arXiv references
> >
>
> Great catch! Our final version will include the correct references.
>
> > Invariance vs. equivariance
> >
>
> Thanks for highlighting this point that wasn’t made clear. Using the language introduced in the original NDF work, the field is equivariant to rigid transformations of the shape, while it's invariant to joint transformation of both the shape and the query point together. We have updated the writing to clarify this point.
>
> > Equation (7). As far as I understand this is not a constrained optimization problem. Phrasing it as a constrained one seems misleading to me.
> >
>
> This is a fair point, the “subject to” is less meant to denote a constraint rather than a substitution of expressions. We will revise to reflect that there are no explicit constraints enforced.
>
> > Lines 166-167. You write “small errors can accumulate”. What is the iterative procedure meant here? Is this the two-step procedure defined by Equations (5) and (6). If so, the word “accumulate” reads a bit weird for a two-step procedure.
> >
>
> Yes, “iterative procedure” does refer to applying Eq. (5) and then applying Eq. (6). This is a fair point that “accumulate” paints a picture of more than just two steps. We have included additional qualitative examples showing how the R-NDF approach can work for three objects via both sequential optimization. In principle this can be extended to even more objects, allowing for a larger and larger opportunity for error buildup. The analogous application of the EBM to the three-object case would refine the solution and hopefully reduce error by having the capacity to “look at” all three objects jointly, rather than one at a time in sequence.
>
> We have also touched up the paper to address the rest of the minor comments, thank you for the detailed feedback.

---

> > ### Comment · Reviewer_1zr8 · 2022-08-24
> > **Response to the authors**
> >
> > Thank you for your response! It addresses my main concerns and I am therefore keeping to my acceptance recommendation.

---

### Official Review · Reviewer_gkTn · 2022-08-01

**Originality:** Good
**Technical Quality:** Very Good
**Clarity Of Presentation:** Good
**Impact:** 3

**Recommendation:**

Weak Accept: I recommend accepting the paper, but will not argue for my recommendation if the majority of other reviewers have a different opinion.

**Summary:**

The paper proposes a learning from demonstration approach based on pretrained category-level dense 3D descriptors (NDFs). The paper formulates single-step object arrangement tasks as a SE(3) transformation between two object instances. The proposed method first encodes SE(3) transformations as a set of point descriptors and then optimizes for the transformation between objects by minimizing the distance in the descriptor space. The method requires a few (10) demonstrations for each task and works for unseen object instances of known categories.

**Issues:**

- Line 71 says that translation equivariance is obtained by mean-centering point clouds. How does that work? Does this assumes the point clouds are complete? A bit more details will be helpful.
- Line 128 mentioned scaling $X_A$ to have scale similar to the salient object parts. How is this scale specified? From what I understand, the objects have various different scales, right?
- How are "task-relevant part of $O_A$" (line 98) specified in the original NDF?
- Why are the EBM model (line 188-202) that learns metrics on the descriptor space not susceptible to overfitting? It seems to be trained on only 10 examples.

**Quality Of The Limitations Section:**

Limitations are addressed clearly

**Reviewer Expertise:**

4: The reviewer is confident but not absolutely certain that the evaluation is correct

**Robotics Focus:**

Sufficient demonstration on hardware

**Strengths And Weaknesses:**

Strengths
- Clever formulation of learning from a few demonstrations.
- Multiple improvements that address the limitation of prior work (NDF).
- Nice real world demo in the supplement.

Weaknesses
- Reliance on object segmentation and complete point cloud.
- Only works for known categories. Implicitly relies on labeled 3D object databases.
- More qualitative examples would improve the presentation, such as showing the variety of objects.

**Summary Of Recommendation:**

The paper's contribution is borderline for me, but I would recommend acceptance based on the novelty and good real-robot demonstration, if not objected by other reviewers.

The paper cleverly leverages category-level descriptors that establishes correspondence between object parts to learn tasks from a few demonstrations and almost no annotation. However, a big underlying assumption is that these category level descriptors can be learned well. I have doubts about these descriptors, as they assume a full correspondence can be established between two object instances from the same category. I don't see how they will work for categories like chairs. I think a descriptor that can deal with partial-to-partial correspondences is crucial to have.

The method also assumes that object rearrangements can be simplified into SE(3) transformations, where in reality many more constraints exist, such as robot kinematics or collisions between object parts. Relying on motion planner is fine for simple scenarios, but eventually the pick and place poses need to consider these factors, especially if objects have more complex geometry and the environment is more clutterred.

---

> ### Author Response · Authors · 2022-08-23
> **Response to Reviewer gkTn**
>
> Thank you for the constructive comments and detailed feedback.
>
> > More qualitative examples … showing the variety of objects.
> >
>
> Good suggestion, we added a new Figure to Appendix A10 showing more shapes. We also added qualitative examples of additional capabilities (see updated Appendix A7-A9).
>
> > assumption that category-level descriptors can be learned well. [these descriptors] assume full correspondence can be established between two object instances. [struggle with] categories like chairs.
> >
>
> Great point, it is an issue if the NDF models have not learned descriptors that encode correspondence properly. Because the NDF [1] encoder produces a global feature vector, descriptor matching on e.g., chairs (where different instances have similar global structure, even with different local structure) might still work. However, we have not tried on categories with varying topology (e.g., chairs), and it’s a fair concern that the NDF might struggle.
>
> Recent approaches in neural implicits learn deformation fields with correction terms [2] and high-dimensional latent spaces [3] to recover dense correspondence for topologically varying shapes. Perhaps these ideas could be incorporated into future versions of the system.
>
> > a descriptor that can deal with partial-to-partial correspondences is crucial
> >
>
> If we understand correctly, this concern mainly relates to the completeness of the point cloud (please correct us if we are mistaken and your comment refers to something else).
>
> Similar to [1], we separated the issue of handling heavily occluded point clouds to focus on improving the multi-object rearrangement. However, we have seen NDF matching work with partial point clouds, as long as they’re not too occluded. Appendix A9 provides an example and discusses this further.
>
> > The method also assumes that object rearrangements can be simplified into SE(3) transformations, where in reality many more constraints exist, such as robot kinematics or collisions
> >
>
> Constraints such as kinematics and collision avoidance are important, and we can in fact incorporate extra costs that take such constraints into account. We included an example of incorporating collision avoidance by adding an extra cost term that penalizes query points that occupy obstacle regions (see updated Appendix A7). Other recent work has shown energy-based modeling is useful for taking robot kinematics into account [4]. This would allow adding additional joint space costs/constraints.
>
> > translation equivariance
> >
>
> This means the translation component of the pose is represented relative to the centroid  of the object point cloud. If we know this centroid, we can subtract it from the 3D points, recover the pose, and add the 3D centroid position back to the translational component. This does not mean the point clouds are complete, it just means that we have a reference point to express the translation relative to. We will update the writing to clarify.
>
> > scaling $\mathcal{X}_{A}$
> >
>
> Thanks for pointing out this that wasn’t clear. The scale is essentially a hyperparameter. The original NDF paper includes ablations showing the query points must be scaled based on the size of the shape (see Table III in [1]). We therefore did not repeat the ablations, and instead tuned the query point scaling to find a value that worked. For real-world objects the scale usually falls between 0.015 and 0.025 (variance on the 3D Gaussian). We have added these details to Appendix A5.2.
>
> > "task-relevant part of OA" specified in the original NDF
> >
>
> [1] used a bounding box around the whole shape.
>
> > EBM overfitting
> >
>
> Please see our general response
>
> > Reliance on object segmentation and complete point cloud… Only works for known categories. relies on labeled 3D objects.
> >
>
> This is a fair point, we do make assumptions that limit the generality and require additional modules. While our main results used relatively complete point clouds, NDFs can work with partial point clouds, as long as the occlusions aren’t too severe (see the updated Appendix A9).
>
> Assuming a known category can be limiting, but it’s also what helps support generalization to similar but unseen shapes. Having a module that focuses on local parts that may be shared across categories is an exciting direction for future work.
>
> Depending on an offline set of 3D objects is another limitation. It would be great to build a method with similar properties that does not depend on ground truth 3D shapes, but we leave this for future work.
>
> [1] Simeonov and Du, et al., “Neural Descriptor Fields: SE(3)-Equivariant Object Representations for Manipulation”, ICRA 2022
>
> [2] Deng, et al., “Deformed Implicit Field: Modeling 3D Shapes with Learned Dense Correspondence”, CVPR 2021
>
> [3] Duggal, et al., “Topologically-Aware Deformation Fields for Single-View 3D Reconstruction”, CVPR 2022
>
> [4] Ganapathi, et al., “Implicit Kinematic Policies: Unifying Joint and Cartesian Action Spaces in End-to-End Robot Learning”, ICRA 2022

---

### Meta-Review · Area_Chair_FCcf · 2022-08-15

**Recommendation:** Accept (Poster)
**Confidence:** 4

**Metareview:**

Below is a summary of the strengths and weaknesses of the paper, according to the reviewers.

Strengths:

- SE(3) equivariance is an interesting and important topic to study.
- There are several improvements over prior work of Neural Descriptor Fields (NDF).
- The method can learn high-precision tasks from just 5-10 demonstrations.
- Experiments show that the method results in improvements over baselines.
- There is a real-world demonstration of the method (although only qualitative results).
- The paper is well written, and the figures are particularly helpful.

Weaknesses:

- The contributions are incremental with respect to NDF.
- The method relies on good object segmentation and a complete point cloud to work.
- The method has several assumptions, such as no distractor objects, a pre-trained NDF for each object, and known object categories.

In the rebuttal, please address the above weaknesses, as well as the other concerns and questions raised by the reviewers.

----------

Update after the rebuttal:

All four reviewers maintained their positive opinion of the paper after the rebuttal. This is a timely paper which proposes another application of implicit neural models to robotics. Reviewers found the paper to be somewhat incremental, but still sufficient for publication at CoRL and with compelling real-world results. It will be a useful paper for anybody working in implicit neural models or dense descriptors, although it may have limited appeal more broadly across the community.

**Best Paper Nomination:**

No

---

> ### Author Response · Authors · 2022-08-23
> **General Response**
>
> **Comment:**
>
> We are grateful for the constructive comments and overall positive feedback from reviewers. Reviewers appreciated the benefits and real-world capabilities enabled by our method (gkTn said we presented “nice real-world demo(s)” and haL1 said that the “real world results are compelling”). We are also happy to hear the reviewers found our paper to be clear (krtE says “the clarity of the paper is good…the figures illustrate the approach well”, haL1 says “the figures are generally helpful”) and that it makes progress toward an important problem (1zr8 points out the paper “makes progress toward a relevant problem”).
>
> **Summary of Additional Experiments, Figures, and Material**
>
> - In response to Reviewer krtE, we have added additional results illustrating how R-NDF may be applied to 3-object rearrangement.
> - In response to Reviewer gkTn and Metareviewer comments, we have added additional results illustrating how R-NDF may be applied even when point clouds are partial.
> - In response to Reviewer gkTn, we illustrate how the NDF framework enables us to incorporate additional constraints beyond descriptor distance matching for SE(3)-transformation inference. In particular, we show we may incorporate the constraint of collision avoidance using the NDF framework, and note that other kinematic constraints could be implemented in a similar manner (and have been implemented in past optimization-based motion planners).
> - We have added a discussion section in the appendix addressing system implementation details (motion planning and collision avoidance), assumptions (segmentation, pretrained NDFs with per-category 3D shape data, etc.), and general limitations as well as additional qualitative visualizations (Reviewer gkTn) .
> - In response to concerns from Reviewer 1zr8 and haL1, we have uploaded a preliminary zip of the code used for the results in the paper. We will further release a fully documented cleaned version of the code upon the public release of the paper.
> - Please see the **attached PDF** for an updated version of the paper.
>
> **Response to general critique of work being “incremental with respect to original NDF”**
>
> While our work builds on and extends prior work on NDFs, we believe that our approach has substantial differences which we discuss below.
>
> 1. In contrast to existing work on NDFs, our approach enables us to manipulate pairs of separately moving objects, in contrast to the “fixed environment” assumption used in the NDF paper. By chaining and composing pairwise relations between objects, we enable our system to generalize to entirely different multi-object scene layouts (as illustrated in Figure 1 and the additional rebuttal experiments).
> 2. Practically implementing NDF alignment between pairs of objects required substantial technical innovation. For example, aligning query points between multiple demonstration objects via variance minimization (as described in Section 4.1) was a non-obvious yet critical design choice.
> 3. Furthermore, we present a novel learning approach with EBMs which enables us to obtain further gains on few-shot demonstration learning.
>
> **Response to common concerns about EBM training (data augmentation to avoid overfitting)**
>
> While we only train our EBM on 10 demonstrations, we find that we are able to still obtain reasonable performance due to large amounts of data augmentation applied to individual point clouds in each demonstration. In particular, we take demonstration point clouds and construct augmented demonstration point clouds by skewing each individual point cloud, adding Gaussian noise, and simulating different occlusion patterns.  Furthermore, note that EBMs are directly learned directly on NDF descriptors — thus our approach, even when overfit, can generalize to novel shapes as long as NDF descriptors are consistent across shape instances. We clarify these data augmentation details in Appendix A2.
>
> **Zip File:**
>
> /attachment/d16285b8397143a98ec7090d945a28865a054090.zip